# Sonic hedgehog signaling in astrocytes mediates cell type-specific synaptic organization

Steven A Hill[1†], Andrew S Blaeser[1†], Austin A Coley[2], Yajun Xie[3], Katherine A Shepard[1], Corey C Harwell[3], Wen-Jun Gao[2], A Denise R Garcia[1,2]*

[1]Department of Biology, Drexel University, Philadelphia, United States; [2]Department of Neurobiology and Anatomy, Drexel University College of Medicine, Philadelphia, United States; [3]Department of Neurobiology, Harvard Medical School, Boston, United States

**Abstract** Astrocytes have emerged as integral partners with neurons in regulating synapse formation and function, but the mechanisms that mediate these interactions are not well understood. Here, we show that Sonic hedgehog (Shh) signaling in mature astrocytes is required for establishing structural organization and remodeling of cortical synapses in a cell type-specific manner. In the postnatal cortex, Shh signaling is active in a subpopulation of mature astrocytes localized primarily in deep cortical layers. Selective disruption of Shh signaling in astrocytes produces a dramatic increase in synapse number specifically on layer V apical dendrites that emerges during adolescence and persists into adulthood. Dynamic turnover of dendritic spines is impaired in mutant mice and is accompanied by an increase in neuronal excitability and a reduction of the glial-specific, inward-rectifying $K^+$ channel Kir4.1. These data identify a critical role for Shh signaling in astrocyte-mediated modulation of neuronal activity required for sculpting synapses.
DOI: https://doi.org/10.7554/eLife.45545.001

*For correspondence:
adg82@drexel.edu

†These authors contributed equally to this work

Competing interests: The authors declare that no competing interests exist.

## Introduction

The organization of synapses into the appropriate number and distribution occurs through a process of robust synapse addition followed by a period of refinement during which excess synapses are eliminated. Failure to establish or maintain appropriate synaptic organization is a hallmark of many neurodevelopmental disorders (*Penzes et al., 2011*). Considerable evidence now shows that, together with neurons, astrocytes are critical regulators of synaptic connectivity and function (*Allen and Eroglu, 2017*; *Adamsky et al., 2018*; *Risher et al., 2014*). Astrocytes interact intimately with synapses to regulate their formation, maturation, and function, and a growing number of astrocyte-secreted proteins that directly mediate synapse formation and elimination have been identified (*Allen et al., 2012*; *Kucukdereli et al., 2011*; *Christopherson et al., 2005*; *Blanco-Suarez et al., 2018*; *Chung et al., 2015*). In addition, astrocytes regulate concentrations of $K^+$ and glutamate in the extracellular space, thereby modulating neuronal activity (*Olsen and Sontheimer, 2008*). Nevertheless, despite the remarkable progress in our understanding of the essential role for astrocytes in regulating synaptic formation and function, the underlying signaling programs mediating astrocyte-dependent regulation of synapse organization remain poorly understood.

The molecular signaling pathway Sonic hedgehog (Shh) governs a broad array of neurodevelopmental processes in the vertebrate embryo, including morphogenesis, cell proliferation and specification, and axon pathfinding (*Fuccillo et al., 2006*; *Ruiz i Altaba et al., 2002*). However, Shh activity persists in multiple cell populations in the postnatal and adult CNS, including progenitor cells, as well as in differentiated neurons and astrocytes (*Traiffort et al., 2010*; *Ahn and Joyner, 2005*;

**eLife digest** A central system of neurons in the spinal cord and brain coordinate most of our body's actions, ranging from regulating our heart rate to controlling our movement and thoughts. As the brain develops, neurons form specialized contacts with one another known as synapses. If the number of synapses is not properly regulated this can disrupt communication between the neurons, leading to diseases like schizophrenia and autism.

As the brain develops, it first forms an excess of synapses and later eliminates unnecessary or weak connections. Various factors, such gene expression or a neuron's level of activity, regulate this turnover process. However, neurons cannot do this alone, and rely on other types of cells to help regulate their behavior. In the central nervous system, for example, a cell called an astrocyte is known to support the formation and activity of synapses. Now, Hill and Blaeser et al. show that astrocytes also exert influence over synaptic turnover during development, leading to long lasting changes in the number of synapses.

Hill, Blaeser et al. revealed that disrupting activity of the signaling pathway known as Sonic hedgehog, or Shh for short, in the astrocytes of mice led to disordered synaptic connections. Notably, neurons produce Shh, suggesting that neurons use this signaling pathway to communicate to specific astrocyte partners. Further experiments showed that reducing astrocyte's ability to respond to Shh impaired synaptic turnover as the brain developed, leading to an overabundance of synapses. Importantly, these effects were only found to influence neuron populations associated with astrocytes that actively use Shh signaling. This suggests that distinct populations of neurons and astrocytes interact in specialized ways to build the connections within the nervous system.

To address how astrocytes use Shh signaling to regulate synaptic turnover, Hill, Blaeser et al. examined gene expression changes in astrocytes that lack Shh signaling. Astrocytes with a reduced capacity to respond to Shh were found to have lower levels of a protein responsible for transporting potassium ions into and out of the cell. This impairs astrocyte's ability to regulate neuronal activity, which may lead to a failure in eliminating unnecessary synapses.

Understanding how synapses are controlled and organized by astrocytes could help identify new ways to treat diseases of the developing nervous system. However, further studies would be needed to improve our understanding of how this process works.

DOI: https://doi.org/10.7554/eLife.45545.002

*Ihrie et al., 2011*; *Harwell et al., 2012*; *Garcia et al., 2010*), where novel and unexpected roles for Shh activity are emerging (*Garcia et al., 2018*). Following injury, Shh has been shown to mitigate inflammation (*Alvarez et al., 2011*; *Allahyari et al., 2019*), and in the cerebellum, Shh derived from Purkinje neurons instructs phenotypic properties of mature Bergmann glia (*Farmer et al., 2016*). In the postnatal cortex, Shh is required for establishing local circuits between two distinct projection neuron populations (*Harwell et al., 2012*). Shh produced by layer V neurons guides the formation of synaptic connections to its layer II/III presynaptic partners, which transduce the Shh signal through non-canonical, Gli-independent mechanisms. We have previously shown that Shh signaling is also active in a discrete subpopulation of cortical astrocytes (*Garcia et al., 2010*), suggesting that Shh signaling mediates both homotypic and heterotypic cellular interactions. Astrocytes engaging in Shh activity are identified by expression of Gli1, a transcriptional effector of canonical Shh signaling (*Fuccillo et al., 2006*). Whether Shh signaling in cortical astrocytes plays a role in synaptic organization of neurons is not known.

In this study, we examined the organization and dynamics of dendritic spines on cortical neurons following selective disruption of Shh signaling in astrocytes. Dendritic spines are the structural hosts of most excitatory synapses and play an important role in the organization and function of neural circuits. We show that deep layer neurons exhibit long-lasting aberrations in the density and turnover of dendritic spines that emerge during postnatal development following selective disruption of Shh signaling in astrocytes. These perturbations in synaptic organization are not observed in upper layer neurons where Gli1 astrocytes are relatively sparse. Chronic in vivo imaging of dendritic spines reveals that mutant mice exhibit lower rates of spine turnover, suggesting impaired structural plasticity. In addition, these mice show a pronounced deficit in expression of the glial-specific inward-

rectifying K$^+$ channel, Kir4.1, as well as an increase in excitability of cortical neurons. Taken together, these data demonstrate that astrocytes act as key modulators of neural activity and structure during postnatal development in a Shh-dependent manner and further establish Shh signaling as a fundamental mediator of synaptic connectivity.

## Results

### Gli1 astrocytes exhibit a distinct laminar distribution in the adult cortex

In the mature cortex, a subpopulation of astrocytes express the transcription factor Gli1, indicating active Shh signaling (*Garcia et al., 2010*). Notably, the distribution of Gli1 astrocytes throughout the cortex is non-uniform, showing a laminar-specific pattern (*Figure 1*). We analyzed the laminar distribution of Gli1 astrocytes in adult *Gli1$^{CreER/+}$*;Ai14 mice, in which tamoxifen administration promotes Cre-mediated recombination of the fluorescent tdTomato reporter protein, permanently marking Gli1-expressing cells. Adult *Gli1$^{CreER/+}$*;Ai14 mice received tamoxifen over three days, and their brains were analyzed two – three weeks later. In our previous study, we observed weak expression of βGal reporter protein in *Gli1$^{CreER/+}$*;*R26$^{lacZ/lacZ}$* mice at time points earlier than 4 weeks after tamoxifen (*Garcia et al., 2010*). In contrast, expression of the tdTomato reporter protein in the Ai14 reporter line shows robust expression as early as two weeks, and a comparable number, distribution and intensity of reporter-positive cells between two and four weeks (*Allahyari et al., 2019*). For these studies, we therefore used a two – three week chase after tamoxifen. The vast majority of marked cells were observed within layers IV and V, with few marked cells observed in layers II/III or VI (*Figure 1*). We performed immunostaining with the pan-astrocytic marker S100β and quantified the fraction of marked astrocytes in cortical layers (*Figure 1*). This analysis showed that 44% of astrocytes in layer IV express Gli1 (*Figure 1*), while 36% of astrocytes in layer V express Gli1. Interestingly, the distribution of Gli1 astrocytes in layer V was not homogenous, showing an enrichment of marked cells in layer Vb, and a distinctive paucity of marked cells in layer Va (*Figure 1*), consistent with the

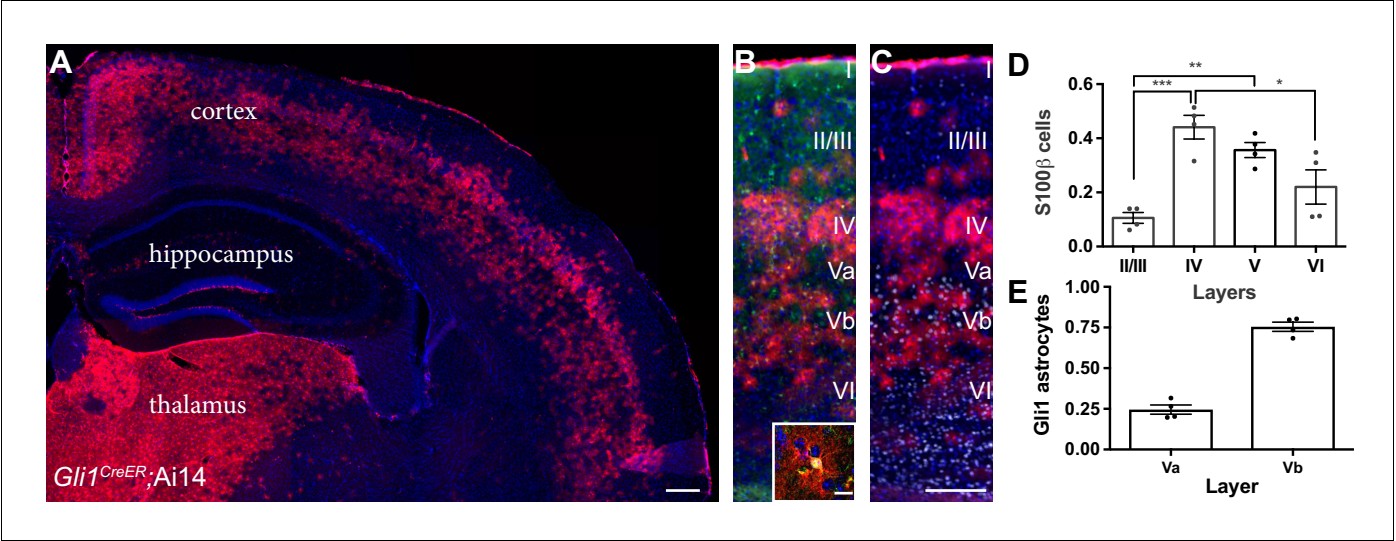

**Figure 1.** Gli1 astrocytes are distributed in a laminar-specific manner. (**A**) Tamoxifen- induced reporter expression in *Gli1$^{CreER/+}$*;Ai14 mice reveals a laminar distribution of cells undergoing active Shh signaling (red). Scale bar, 250 µm. (**B**) Gli1+ cells (red) are a subset of all astrocytes (S100β, green, see inset; scale bar, 12.5 µm). (**C**) Layer analysis of Gli1 astrocytes (red) with DAPI and the deep layer marker Ctip2 (gray) illustrate their layer-specific organization. Scale bar, 125 µm. (**D**) The fraction of astrocytes expressing Gli1 in each cortical layer. Data points indicate individual animals, bars represent mean ± SEM. Statistical significance was assessed by one-way ANOVA with Tukey's post hoc test for multiple comparisons. *, p<0.05; **, p<0.01; ***, p<0.001. (**E**) The fraction of Gli1 astrocytes in sublayers Va and Vb. Bar graphs represent mean value ± SEM. n = 4 animals.

DOI: https://doi.org/10.7554/eLife.45545.003

The following source data is available for figure 1:

**Source data 1.** Distribution of Gli1 astrocytes in cortical layers.

DOI: https://doi.org/10.7554/eLife.45545.004

localization of Shh producing neurons in layer Vb (*Harwell et al., 2012*; *Garcia et al., 2010*). We analyzed the fraction of Gli1 astrocytes in each sublayer of layer V and found that 75% of marked cells were found in layer Vb whereas only 25% were found in layer Va. Layers II/III and VI showed the lowest fractions of marked astrocytes, with only 11% and 22% of S100β cells expressing tomato, respectively. These data demonstrate that Gli1 astrocytes are preferentially localized in deep cortical layers and suggest their interactions with local neurons may differ from that of Gli1-negative astrocytes in superficial layers.

## Shh signaling is required for cell type-specific synaptic connectivity

Transduction of canonical Shh signaling begins with binding of Shh to its transmembrane receptor, Patched1 (Ptch), relieving inhibition of a second, obligatory co-receptor Smoothened (Smo). To investigate the role of Shh signaling in astrocyte function, we performed conditional knockout of *Smo* selectively in astrocytes using *Gfap-Cre* transgenic mice in which Cre expression is regulated by the full-length mouse *Gfap* promoter (*Garcia et al., 2004*) (*Gfap Smo* CKO). Because *Gfap*-mediated Cre recombination targets astrocyte progenitors expressing GFAP which are present at birth, this is an effective tool for targeting the cortical astrocyte population for selective gene deletion. We validated the Cre-mediated recombination pattern in *Gfap-Cre* mice by crossing them with Ai14 reporter mice (*Gfap-Cre*;Ai14), that express the tdTomato reporter protein. Although individual floxed alleles possess distinct recombination efficiencies, owing to specific characteristics of each allele including distance between loxP sites or their accessibility due to chromatin structure (*Vooijs et al., 2001*), analysis of recombination in a reporter line provides a useful approximation of the expression pattern of a given Cre driver. We performed single cell analysis of the identity of recombined cells. The vast majority of tomato-positive cells showed a bushy morphology, typical of protoplasmic astrocytes (*Figure 2—figure supplement 1*). Single cell analysis of double staining with S100β showed that nearly 70% were co-labeled, identifying them as astrocytes. We also identified a small fraction (6%) of tomato cells as oligodendrocytes. A small fraction (9%) of tomato cells were co-labeled with the neuronal marker, NeuN, but this represented only 1.3% of cortical neurons (*Figure 2—figure supplement 1*). Importantly, although a minor fraction of recombined cells was identified as neurons or oligodendrocytes, nearly all cortical astrocytes analyzed expressed the tdTomato reporter protein (95%; *Figure 2—figure supplement 1*), suggesting effective targeting of the cortical astrocyte population using this *Gfap-Cre* driver.

We performed qPCR on whole cortex from *Gfap Smo* CKO mice and littermate controls and found a 50% reduction in the number of *Smo* transcripts (*Figure 2—figure supplement 2*). As *Smo* is also expressed in neurons (*Harwell et al., 2012*), the remaining *Smo* transcripts are likely due to neurons that do not undergo Cre-mediated recombination in these mice. Importantly, *Gfap Smo* CKO mice show a nearly complete loss of Gli1 activity, with no difference in the number of astrocytes in the mature cortex, demonstrating effective interruption of canonical Shh signaling (*Garcia et al., 2010*). These data demonstrate that this *Gfap-Cre* driver is an effective tool for selective deletion of *Smo* in cortical astrocytes, and that canonical, Gli-mediated Shh activity is effectively abolished.

During postnatal development, Shh is required for establishing synaptic connectivity between layer V neurons and their presynaptic partners in layer II/III, which is mediated by Gli-independent, non-canonical signaling (*Harwell et al., 2012*). To examine whether canonical, Gli-mediated Shh signaling in astrocytes plays a role in establishing synaptic connectivity of cortical neurons, we evaluated spine density of apical dendrites on layer V neurons in the somatosensory cortex of *Gfap Smo* CKO mice across postnatal development. Dendritic spines receive the vast majority of excitatory input, thereby serving as a useful readout of synaptic connectivity (*Nimchinsky et al., 2002*). To visualize neurons for reliable identification and tracing, *Gfap Smo* CKO mice were crossed with Thy1-GFPm transgenic mice, which sparsely express GFP in a subset of layer V cortical neurons (*Feng et al., 2000*). Dendritic spines undergo a period of dynamic reorganization during postnatal development during which there is an initial overproduction of spines over the first 2–3 weeks of postnatal development followed by a period of synaptic pruning that refines the precise connectivity of cortical circuits (*Zuo et al., 2005a*). At P14, spine density was comparable between *Gfap Smo* CKO mice and wild type (WT) littermate controls, suggesting that early synaptogenesis does not require astrocytic Shh signaling (0.58 ± 0.03 spines/μm and 0.56 ± 0.01 spines/μm, WT [$n = 7$ cells] and *Gfap Smo* CKO [$n = 6$ cells], respectively, p=0.64, two animals per genotype). Between P14 and

P21, spine density remained stable in WT, but showed a dramatic increase at P28 (*Figure 2*). This period of synapse addition was followed by a steady reduction in spine density at P42 and into adulthood (≥P90), reflecting the developmental elimination of spines. In contrast, *Gfap Smo* CKO mice showed an accelerated timeline of spine addition in which spine density increased dramatically between P14 and P21 (*Figure 2*). Spine density was significantly higher at P21 in *Gfap Smo* CKO mice compared to controls (0.53 ± 0.05 spines/μm and 0.76 ± 0.06 spines/μm, WT [*n* = 9 cells] and *Gfap Smo* CKO [*n* = 9 cells], respectively, p=0.008, three animals per genotype). Although there was a modest reduction in spine density as animals matured, spine density remained elevated in adult *Gfap Smo* CKO mice compared to WT controls (0.36 ± 0.02 spines/μm and 0.60 ± 0.02 spines/μm in WT [*n* = 24 cells] and *Gfap Smo* CKO [*n* = 23 cells], respectively, p<0.0001, four animals per

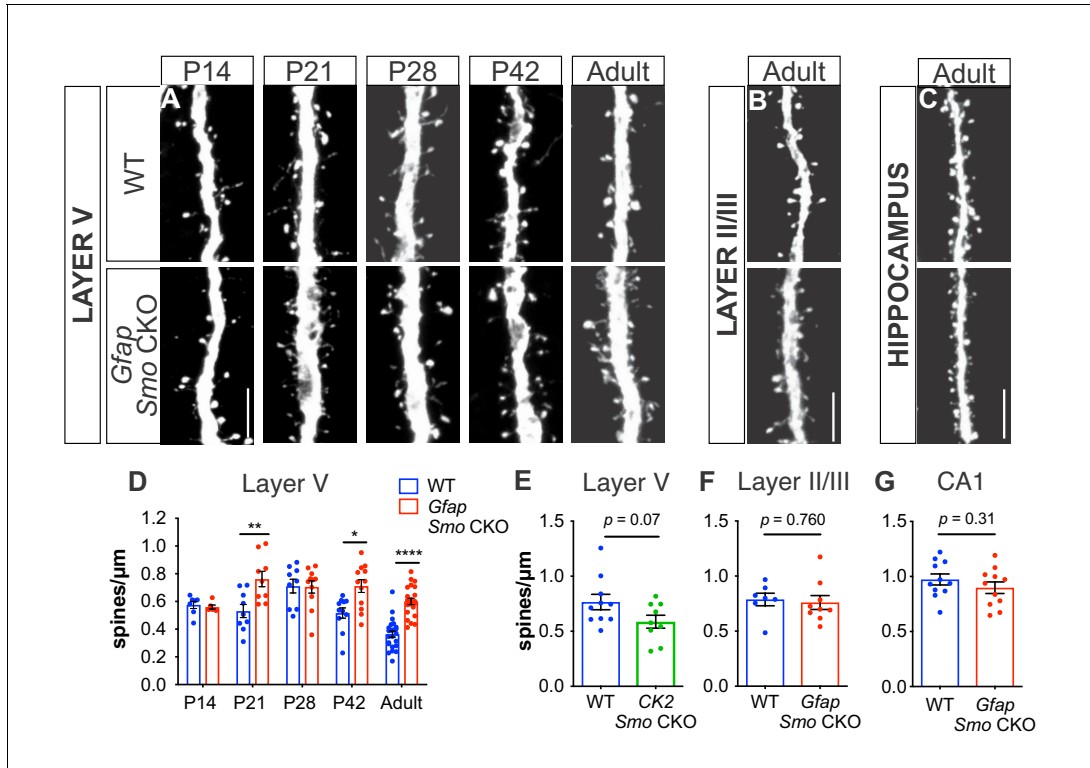

**Figure 2.** Loss of Shh signaling in astrocytes results in increased spine density on layer V neurons. (A-C) GFP immunostaining of representative apical dendritic segments from (A) layer V neurons, (B) layer II/III neurons, and (C) CA1 neurons in WT and *Gfap Smo* CKO mice. Scale bar, 5 μm. (D) Spine density of apical dendrites from layer V neurons across various postnatal ages and in adult mice. Data points represent individual cells, bars represent mean ± SEM. Statistical analysis performed by two-way ANOVA with Tukey's post-hoc test for multiple comparisons. *, p<0.05; **, p<0.01; ****, p<0.0001. Significance is stated as *Gfap Smo* CKO compared to WT at a given age. (E) Spine density of apical dendrites from layer V neurons in adult *CamKIIa Smo* CKO mice. (F–G) Spine density of apical dendrites from layer II/III (F) and CA1 pyramidal neurons (G) in WT and *Gfap Smo* CKO mice. Data points represent individual cells, bars represent mean ± SEM. Statistical significance assessed by Student's t-test. *n* ≥ 3 animals per genotype for all ages except P14, where *n* = 2 animals per genotype.

DOI: https://doi.org/10.7554/eLife.45545.005

The following source data and figure supplements are available for figure 2:

**Source data 1.** Analysis of spine density.
DOI: https://doi.org/10.7554/eLife.45545.010
**Figure supplement 1.** *Gfap-Cre* recombination occurs primarily in astrocytes.
DOI: https://doi.org/10.7554/eLife.45545.006
**Figure supplement 2.** *Gfap Smo* CKO mice have reduced levels of *Smo*.
DOI: https://doi.org/10.7554/eLife.45545.007
**Figure supplement 2—source data 1.** Quantification of *Smo* mRNA.
DOI: https://doi.org/10.7554/eLife.45545.008
**Figure supplement 3.** Spine density on layer V basal dendrites are not affected in adult *Gfap Smo* CKO mice.
DOI: https://doi.org/10.7554/eLife.45545.009

genotype; *Figure 2*). Thus, whereas WT mice experience a 49% reduction in spine density from the peak of spine density at P28 to adulthood, *Gfap Smo* CKO mice exhibit a 22% reduction from their peak at P21 to adulthood. Interestingly, we find no difference in basal dendrite spine density in *Gfap Smo* CKO mice (0.60 ± 0.04 spines/um and 0.62 ± 0.04 spines/μm in WT [*n* = 10 cells] and *Gfap Smo* CKO [*n* = 10 cells], respectively, p=0.71, three animals per genotype; *Figure 2—figure supplement 3*). This is in contrast to embryonic deletion of Shh from neurons using the Emx-Cre driver, which produces a reduction in spine density of basal dendrites on layer V neurons (*Harwell et al., 2012*). These data suggest that while the early stages of synapse formation proceed independently of astrocytic Shh signaling, the maturing cortical circuit requires intact Shh activity in astrocytes for the developmental pruning of excess synapses necessary to achieve its mature organization.

To confirm that elevated spine density was due specifically to a loss of astrocytic Shh signaling, we interrogated the spine density of pyramidal cells in two different regions where Gli1 astrocytes are relatively sparse (see *Figure 1*). Pyramidal neurons in layer II/III of WT mice showed a higher spine density than layer V neurons (0.79 ± 0.06 spines/μm, *n* = 7 cells, four animals) consistent with previous studies (*Holtmaat et al., 2005*). However, this was not significantly different from the spine density observed in *Gfap Smo* CKO mice (0.76 ± 0.06 spines/μm, p=0.76, *n* = 9 cells, five animals; *Figure 2*). We also analyzed pyramidal neurons in the hippocampus. Although the dentate gyrus of the hippocampus harbors a population of Gli1-expressing adult neural progenitor cells, mature, differentiated astrocytes expressing Gli1 are relatively sparse (*Ahn and Joyner, 2005*; *Han et al., 2008*) (*Figure 1*). The spine density of CA1 pyramidal neurons in adult mice was higher than in cortical neurons, consistent with previous studies (*Attardo et al., 2015*; *Perez-Cruz et al., 2011*). However, there was no significant difference in spine density between *Gfap Smo* CKO mice and WT controls (0.97 ± 0.05 spines/μm and 0.90 ± 0.05 spines/μm in WT [*n* = 11 cells] and *Gfap Smo* CKO [*n* = 11 cells], respectively, p=0.32, three animals per genotype; *Figure 2*). Our analysis of the recombination pattern of the *Gfap-Cre* driver showed that some cortical neurons undergo recombination (*Figure 2—figure supplement 1*). To rule out the possibility that *Smo* deletion in a small population of neurons is responsible for the elevated spine density of layer V neurons, we deleted *Smo* in excitatory neurons using the *CamKIIα-Cre* driver (*CK2 Smo* CKO), and evaluated spine density of layer V neurons in adult mice. We found no significant difference in spine density between *CK2 Smo* CKO mice and littermate controls (0.77 ± 0.07 spines/μm and 0.59 ± 0.06 spines/μm in WT [*n* = 10 cells] and *CK2 Smo* CKO [*n* = 9 cells], respectively, p=0.07, three animals per genotype; *Figure 2*). Taken together, these data suggest that Shh activity in astrocytes is necessary for the developmental reorganization of dendritic spines specifically on layer V cortical neurons. Notably, we observed elevated spine density along the apical dendrites of layer V neurons, which traverse layer II/III, where few Gli1 astrocytes are found. This suggests that the modification of synapse organization mediated by Gli1 astrocytes does not occur through direct astrocyte-synapse interactions.

## Synapses are more stable in *Gfap Smo* CKO mice

Dendritic spines are dynamic structures that undergo rapid formation and elimination during postnatal cortical development. As cortical circuits mature, the fraction of spines undergoing dynamic turnover declines concomitant with an increase in spine stability (*Zuo et al., 2005a*; *Holtmaat et al., 2005*). The dynamic turnover of spines has long been considered a structural correlate of synaptic plasticity and can be regulated in an activity-dependent manner (*Lendvai et al., 2000*; *Trachtenberg et al., 2002*; *Holtmaat and Svoboda, 2009*; *Zuo et al., 2005b*). To evaluate the role of astrocytic Shh signaling in mediating spine dynamics, we performed repeated in vivo imaging of the apical tufts of layer V neurons in the somatosensory cortex through a cranial window (*Holtmaat et al., 2009*). We confirmed the layer V identity of imaged neurons by following individual dendritic segments to their soma and created 3D reconstructions of imaged neurons. Only dendrites with soma in layer V, typically 500 μm – 600 μm from the surface of the brain, were analyzed. In Thy1-GFPm mice, expression of GFP is developmentally regulated such that at P14, the intensity of GFP expression and the density of fluorescently labeled neurons within the 3 mm cranial window is insufficient for reliable imaging, as has been previously reported (*Holtmaat et al., 2005*). However by P17, GFP expression was sufficiently dense and bright to enable reliable imaging. We analyzed the fraction of spines undergoing dynamic turnover over 2 days in young mice at P17-P21 and P28-P32. In WT mice, the turnover ratio was 0.14 at P17-P21 (*n* = 5 mice) and 0.10 in P28-P32 mice (*n* = 3 mice; *Figure 3*). In *Gfap Smo* CKO mice however, the turnover ratio was lower at P17-P21

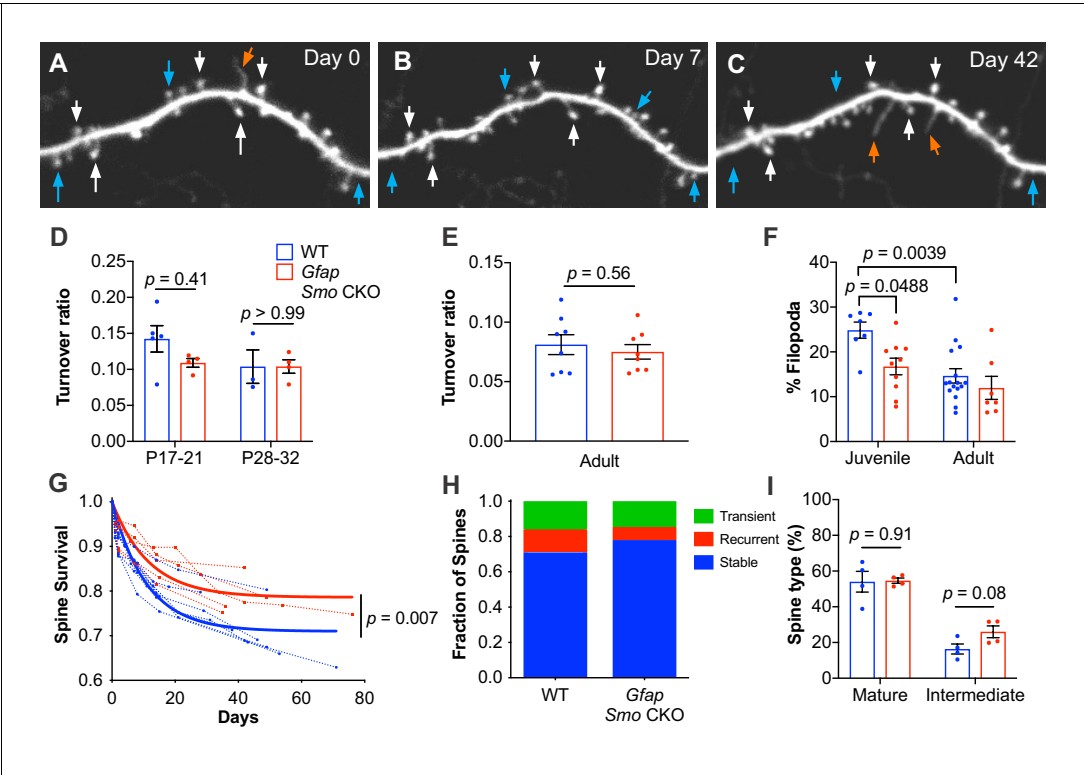

**Figure 3.** Loss of astrocytic Shh signaling impairs structural plasticity. (A–C) Example dendrite segment imaged repeatedly over 6 weeks. Day 0 indicates the first day of imaging, subsequent imaging days indicated. White arrows, stable spines; cyan arrows, transient spines; orange arrows, filopodia. (D) Turnover ratios in WT ($n$ = 3–5 animals per age group) and *Gfap Smo* CKO juvenile mice ($n$ = 4 animals per age group) analyzed over 2 days. (E) Turnover ratios in WT ($n$ = 8 animals) and *Gfap Smo* CKO ($n$ = 8 animals) adult mice analyzed over 7 days. Statistical analysis by two-way ANOVA, Tukey's post hoc test (D) and unpaired Student's t-test (E). (F) Fraction of protrusions identified as filopodia in juvenile (WT, $n$ = 7 animals; *Gfap Smo* CKO, $n$ = 10 animals) and adult mice (WT, $n$ = 16: *Gfap Smo* CKO, $n$ = 7 animals). Statistics by two-way ANOVA, Tukey's post hoc test. (G) Comparison of spine survival curves for WT and *Gfap Smo* CKO neurons. Each dashed curve represents the curve from an individual mouse (WT, $n$ = 7 animals; *Gfap Smo* CKO, $n$ = 5 animals). Solid curves represent best-fits to exponential decay models. (H) The fraction of spines belonging to the stable, recurrent, and transient populations. Statistical significance was assessed by Student's t-test for each class (stable, recurrent, or transient; $n$ = 7 and 8 animals, WT and *Gfap Smo* CKO, respectively). Stable, p=0.0485; Recurrent, p=0.1023; Transient, p=0.7709. (I) Breakdown of spine morphology in juvenile mice. Statistical analysis by unpaired Student's t-test for each spine class. For graphs D-F and I, data points represent individual animals, bars represent mean ± SEM.

DOI: https://doi.org/10.7554/eLife.45545.011

The following source data, source code and figure supplements are available for figure 3:

**Source data 1.** Spine turnover data.
DOI: https://doi.org/10.7554/eLife.45545.014
**Source code 1.** MATLAB-based custom-written code for longitudinal analysis of spine dynamics.
DOI: https://doi.org/10.7554/eLife.45545.015
**Figure supplement 1.** Classification of protrusions.
DOI: https://doi.org/10.7554/eLife.45545.012
**Figure supplement 2.** Screenshot depicting the graphical user interface used for longitudinal tracking of spines across multiple days of imaging.
DOI: https://doi.org/10.7554/eLife.45545.013

than in WT mice, and remained stable at P28-P32 (0.11 and 0.10, respectively, $n$ = 4 mice; *Figure 3*). This suggests that dendritic spines in *Gfap Smo* CKO mice stabilize earlier than those in WT mice. Consistent with this, we found that the fraction of filopodia was lower in juvenile *Gfap Smo* CKO mice compared to WT controls (*Figure 3*). Filopodia are structural precursors of dendritic spines and exhibit higher rates of dynamic turnover than mature spines (*Ziv and Smith, 1996*; *Berry and Nedivi, 2017*). Accordingly, the density of filopodia declines as the cortex matures (*Zuo et al., 2005a*; *Grutzendler et al., 2002*). Indeed, the fraction of all protrusions in juvenile WT mice ($n$ = 7 mice) identified as filopodia was 25%, but that fraction declined significantly to 15% in adults ($n$ = 16

mice, p=0.0039; *Figure 3*). In contrast, we did not observe this age-dependent decline in *Gfap Smo* CKO mice (17% and 12% in juvenile [*n* = 10 mice] and adult [*n* = 7 mice] *Gfap Smo* CKO mice, respectively, p=0.398). The filopodial fraction in juvenile *Gfap Smo* CKO mice was already significantly below WT levels and comparable to adult WT levels, suggesting that nascent spines undergo accelerated stabilization in the absence of astrocyte-mediated Shh signaling.

We performed further analysis of spine morphologies and classified spines as mature or intermediate, to determine if there is an increase in mature spines. Protrusions with a mushroom-like morphology were classified as mature, and thin protrusions lacking a discernible spine head, or exhibiting a relatively dim, small head were identified as intermediate spines (*Figure 3—figure supplement 1*). In P17-21 mice, WT and CKO mice showed similar proportions of mature spines (54% and 55% in WT and CKO, respectively, *n* = 4 mice per genotype; *Figure 3*), however *Gfap Smo* CKO mice showed an apparent increase in protrusions with an intermediate morphology (16% and 26% in WT and CKO, respectively; *Figure 3*), though this was not statistically significant. This suggests that these intermediate protrusions reflect transitional morphologies as spines undergo maturation, and become more stable. Consistent with this, adult *Gfap Smo* CKO mice trended toward a modest increase in the fraction of mature spines, compared to their WT controls (55% and 65% in WT and CKO, respectively, *n* = 4 mice per genotype; *Figure 3—figure supplement 1*), whereas the fraction of intermediate spines was comparable at this age (27% and 23% in WT and CKO, respectively).

It is well established that the rate of synaptic turnover declines considerably as animals mature, reflecting an increase in stability of synaptic connections (*Zuo et al., 2005a*; *Holtmaat et al., 2005*). We next sought to investigate the long-term stability of individual spines in adult mice by imaging dendrites weekly for up to 6 weeks. We investigated the fraction of spines identified on the first day of imaging that persisted in subsequent imaging sessions using custom written MATLAB code (*Figure 3—figure supplement 2*). We calculated the survival fraction by fitting to an exponential decay model. This analysis revealed a larger proportion of long-lived stable spines in *Gfap Smo* CKO mice compared to WT control (71% and 79% in WT [*n* = 7 mice] and *Gfap Smo* CKO [*n* = 5 mice], respectively, p=0.0068, Extra sum-of-squares F test; *Figure 3*), suggesting an increase in stability of dendritic spines. Interestingly, among the population of dynamic spines, we found two distinct populations consisting of transient spines, which disappeared and did not reappear for the duration of the study, and recurrent spines, which disappeared and then subsequently reappeared. Our data showed that the shift towards increased spine stability in the *Gfap Smo* CKO neurons was entirely due to a reduction in the proportion of recurrent spines, with nearly identical fractions of transient spines observed across both genotypes (*Figure 3*). These data suggest that structural plasticity is impaired in *Gfap Smo* CKO mice. This deficit in structural plasticity emerges during the third week of postnatal development and persists into adulthood. Taken together, these data demonstrate that astrocyte-mediated Shh signaling is required for establishing and maintaining the organization of cell type specific cortical synapses.

## Shh signaling regulates Kir4.1 expression in cortical astrocytes

In both the developing and adult CNS, astrocytes directly eliminate synapses in an activity-dependent manner through MEGF10 and MERTK, two phagocytic receptors enriched in astrocytes (*Chung et al., 2013*). To investigate whether these genes are negatively regulated in *Gfap Smo* CKO mice, we examined their expression in the cortex of adult mice by quantitative PCR (qPCR). Expression of both *Mertk* and *Megf10* was comparable between WT and *Gfap Smo* CKO mice (dCq values, *Mertk*: 5.97 ± 0.23 and 5.87 ± 0.06 in WT [*n* = 3 mice] and *Gfap Smo* CKO [*n* = 6 mice], respectively, p=0.54; *Megf10:* 8.02 ± 0.38 and 7.69 ± 0.18 in WT [*n* = 3 mice] and *Gfap Smo* CKO [*n* = 6 mice], respectively, p=0.39; *Figure 4—figure supplement 1*). These data suggest that direct engulfment of synapses by astrocytes through these pathways is not regulated by Shh signaling and is not responsible for the increased spine density seen in *Gfap Smo* CKO animals.

In the mature cerebellum, Shh signaling between Purkinje neurons and Bergmann glia regulates expression of the glial specific, inward-rectifying K$^+$ channel, Kir4.1 (*Farmer et al., 2016*). In the cortex, the distribution of Kir4.1 exhibits a laminar pattern similar to that of Gli1 astrocytes, such that expression is enriched in layers IV and Vb, and reduced in layer Va (*Figure 4*). High resolution, confocal analysis showed that Kir4.1 is localized in the processes of Gli1 astrocytes (*Figure 4*). Kir4.1 expression is localized to astrocytic endfeet and is found surrounding neuronal somata in the spinal

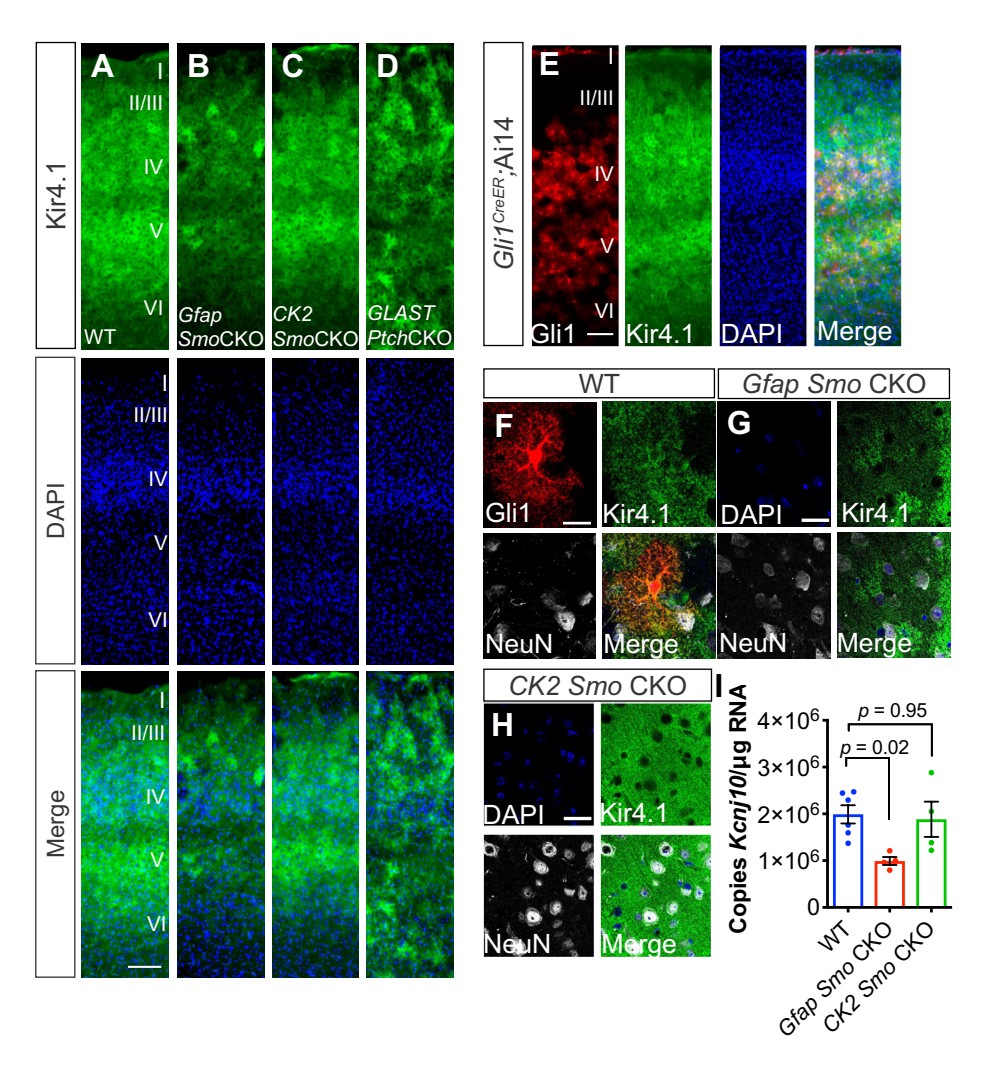

**Figure 4.** Shh signaling regulates Kir4.1 expression in the cortex. (A–D) Immunofluorescence for Kir4.1 in the cortex of adult WT (A), *Gfap Smo* CKO (B), *CK2 Smo* CKO (C) and GLAST Ptch CKO mice (D). Scale bar, 125 μm. (E) Fluorescence micrographs showing the distribution of Gli1 astrocytes (red) and Kir4.1 (green) in the cortex of an adult *Gli1CreER/+*;Ai14 mouse counterstained with DAPI (blue). (F–G) Confocal images of Gli1 (F, red) Kir4.1 (green) and NeuN (white) from adult WT (F), *Gfap Smo* CKO (G), or *CamK2a Smo* CKO (H) mice. Scale bar, 25 μm. (I) Gene expression levels in the cortex of *Kcnj10* from WT (*n* = 6 animals) and *Gfap Smo* CKO or *CK2 Smo* CKO mice (*n* = 4 each). Data points represent individual animals and bar graphs represent mean value ± SEM. Statistical significance was assessed by one-way ANOVA with Tukey's test for multiple comparisons. Significance stated on the graph is compared to WT.

DOI: https://doi.org/10.7554/eLife.45545.016

The following source data and figure supplement are available for figure 4:

**Source data 1.** Measurement of gene expression.
DOI: https://doi.org/10.7554/eLife.45545.018
**Figure supplement 1.** Quantitative PCR analysis of known spine and synapse modulators shows no change in expression in *Gfap Smo* CKO mice.
DOI: https://doi.org/10.7554/eLife.45545.017

cord and brain (*Kelley et al., 2018*; *Cui et al., 2018*; *Higashi et al., 2001*). In WT mice, expression of Kir4.1 surrounded many NeuN-positive neuronal somata in layer V (*Figure 4*). In *Gfap Smo* CKO mice, however, there was a pronounced reduction in the expression of Kir4.1 throughout the cortex, and peri-somal Kir4.1 expression was severely diminished (*Figure 4*). In order to quantify expression

levels of Kir4.1, we measured transcript abundance in the cortex of *Gfap Smo* CKO mice and WT controls by droplet digital PCR (ddPCR). In *Gfap Smo* CKO mice, there was a 50% reduction in *Kcnj10* transcripts compared to WT controls ($1.99 \times 10^6 \pm 1.93 \times 10^5$ copies/µg and $9.92 \times 10^5 \pm 8.49 \times 10^4$ copies/µg in WT [$n = 6$ mice] and *Gfap Smo* CKO [$n = 4$ mice], respectively, p=0.02). There was no difference in *Kcnj10* levels in *CK2 Smo* CKO mice compared to WT ($1.88 \times 10^6 \pm 3.77 \times 10^5$ copies/µg, p=0.95, $n = 4$ mice; *Figure 4*), suggesting that downregulation of Kir4.1 is due to astrocyte-specific, and not neuronal, disruption of Shh signaling. Conversely, *Glast-CreER*-mediated deletion of the Shh receptor *Ptch1*, a negative regulator of Shh signaling, dramatically upregulated Kir4.1 expression in the cortex (*Glast Ptch* CKO; *Figure 4*). Kir4.1 expression is associated with glutamate uptake both in vitro and in vivo (*Kucheryavykh et al., 2007*; *Djukic et al., 2007*; *Tong et al., 2014*). We therefore measured expression levels of the astrocyte-specific glutamate transporters *Glast* and *Glt1*. There was no difference in expression of these genes in *Gfap Smo* CKO mice compared to WT (*Glt1*: $1.08 \times 10^7 \pm 6.6 \times 10^5$ copies/µg and $9.58 \times 10^6 \pm 2.1 \times 10^5$ copies/µg in WT and *Gfap Smo* CKO, respectively, p=0.15, $n = 3$ mice per genotype; *Glast*: $7.4 \times 10^6 \pm 4.4 \times 10^5$ copies/µg and $6.7 \times 10^6 \pm 6.5 \times 10^5$ copies/µg in WT and *Gfap Smo* CKO, respectively, p=0.41, $n = 3$ mice per genotype; *Figure 4—figure supplement 1*). It should be noted that Kir4.1 was not restricted to Gli1 astrocytes in layers IV and Vb. Nevertheless, single transcript quantification of *Kcnj10* transcripts in *Gfap Smo* CKO show a pronounced reduction, suggesting that Kir4.1 expression in cortical astrocytes is regulated, in part, by Shh signaling.

## Neurons exhibit hyperexcitability in *Gfap Smo* CKO mice

To examine whether neurons in *Gfap Smo* CKO mice exhibit disruptions in physiological activity, we performed whole-cell patch clamp recordings of layer V pyramidal neurons from coronal brain slices at P21, and examined action potential firing and membrane properties. Using current clamp, we recorded action potential spikes per injected current at multiple step currents from −300 pA to +650 pA. Our results revealed a significant increase in spike numbers with high current injections (>500 pA) and total overall spikes in *Gfap Smo* CKO animals compared to WT control ($326 \pm 54$ and $574 \pm 71$ spikes in WT [$n = 11$ cells from four animals] and *Gfap Smo* CKO [$n = 7$ cells from three animals], respectively, p=0.01; *Figure 5*). We also measured other membrane properties, including input resistance, resting membrane potential and tau, and found no significant difference between *Gfap Smo* CKO and WT neurons (*Figure 5—figure supplement 1*). However, we observed a significant decrease in action potential threshold ($−30.6 \pm 2.0$ and $−37.9 \pm 2.0$ mV in WT and *Gfap Smo* CKO, respectively, p=0.0441), accompanied by an increase in action potential amplitude ($59.6 \pm 2.6$ and $73.3 \pm 3.0$ pA in WT and *Gfap Smo* CKO, respectively, p=0.0083), and a trending decrease in action potential ½ width ($1.0 \pm 0.92$ and $0.76 \pm 0.02$ ms in WT and *Gfap Smo* CKO, respectively, p=0.05) in *Gfap Smo* CKO neurons, consistent with an increase in neuron excitability (*Figure 5*). Together, these results suggest *Gfap Smo* CKO neurons exhibit an increase in neuronal excitability consistent with excess extracellular $K^+$.

To examine excitatory synaptic transmission, we recorded spontaneous and miniature excitatory postsynaptic currents (sEPSCs and mEPSCs) in layer V pyramidal neurons. We found an increase in both the frequency ($0.74 \pm 0.08$ and $1.18 \pm 0.19$ Hz in WT [$n = 16$ cells] and *Gfap Smo* CKO [$n = 16$ cells], respectively, p=0.042) and amplitude ($8.54 \pm 0.38$ and $11.33 \pm 0.91$ pA in WT and *Gfap Smo* CKO, respectively, p=0.008) of sEPSCs in slices from *Gfap Smo* CKO mice compared to WT control slices (four animals per genotype; *Figure 5*). Furthermore, we observed an increase in the frequency ($0.40 \pm 0.05$ and $1.10 \pm 0.23$ Hz in WT and *Gfap Smo* CKO, respectively, p=0.003) and amplitude ($7.54 \pm 0.57$ and $10.34 \pm 0.74$ pA in WT and *Gfap Smo* CKO, respectively, p=0.005) in mEPSCs in the presence of tetrodotoxin (TTX), indicating an increase in postsynaptic response of neurons in *Gfap Smo* CKO mice ($n = 14$ and 11 cells from WT and *Gfap Smo* CKO, respectively, four animals per genotype; *Figure 5*). These data suggest that loss of Shh activity in astrocytes produces disturbances in both pre and postsynaptic activity. Moreover, these data suggest that astrocytes modulate both neuronal excitability and excitatory synaptic transmission in a Shh-dependent manner.

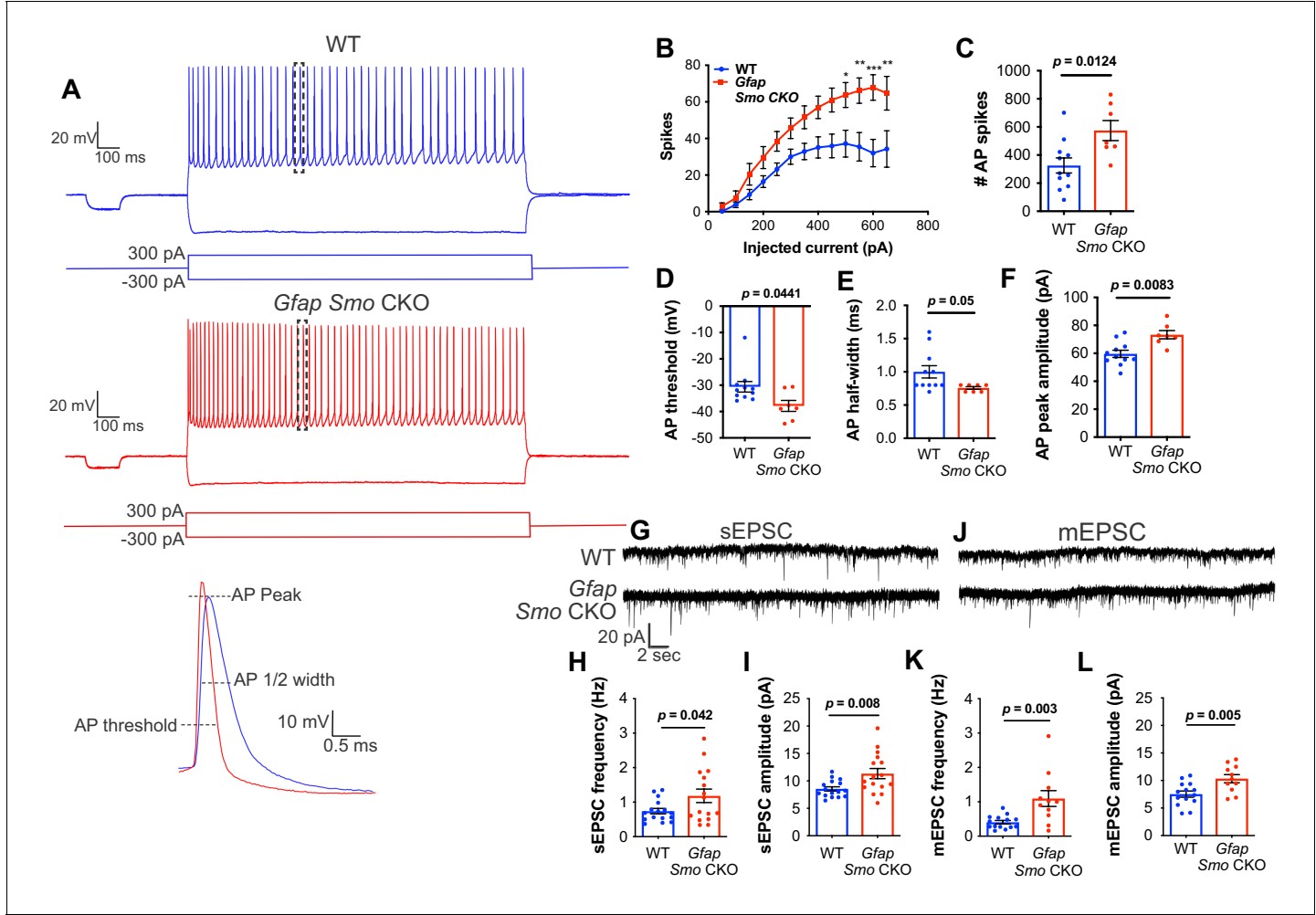

**Figure 5.** Neurons in *Gfap Smo* CKO mice exhibit a significant increase in excitability and synaptic transmission. (**A**) Example traces of action potentials from layer V pyramidal neurons in WT and *Gfap Smo* CKO mice at P21. Samples of action potential spikes (dashed lines, lower panel) describe AP threshold, AP peak amplitude, and AP 1/2 width of pyramidal neurons. (**B**) Line graph displays relationship between action potential spike numbers (y-axis) and current injection; *Gfap Smo* CKO neurons exhibit an increase in AP spikes (>500 pA) compared to WT neurons. Statistical significance was assessed by two-way ANOVA with Sidak's test for multiple comparisons and stated as *Gfap Smo* CKO compared to WT at a given current (n = 11 and 7 cells from WT and *Gfap Smo* CKO, respectively). (**C–F**) Bar graphs describe an increase in total AP spike numbers (**C**), reduction in AP threshold (mV) (**D**), reduction in AP 1/2 width (**E**), and increase in AP peak amplitude (pA) (**F**) in *Gfap Smo* CKO neurons. (**G**) Example traces of sEPSCs from layer V pyramidal neurons recorded in the presence of picrotoxin from WT and *Gfap Smo* CKO mice at P21. (**H–I**) Summary graphs of sEPSC frequency and amplitude (n = 14 and 11 cells from WT and *Gfap Smo* CKO animals, respectively). (**J**) Example traces of mEPSCs from layer V pyramidal neurons recorded in the presence of picrotoxin and TTX. (**K–L**) Summary graphs of mEPSC frequency and amplitude (n = 16 cells per genotype). *Gfap Smo* CKO neurons exhibit increases in both frequency and amplitude in sEPSCs and mEPSCs recordings. Statistical significance was assessed by unpaired Student's t-test. For each graph, data points represent individual cells. All data are from at least three animals per genotype.

DOI: https://doi.org/10.7554/eLife.45545.019

The following source data and figure supplement are available for figure 5:

**Source data 1.** Physiological properties of neurons and response to current injections.
DOI: https://doi.org/10.7554/eLife.45545.021

**Source data 2.** Measurements of spontaneous and miniature EPSCs.
DOI: https://doi.org/10.7554/eLife.45545.022

**Figure supplement 1.** Membrane properties of *Gfap Smo* CKO layer V pyramidal neurons.
DOI: https://doi.org/10.7554/eLife.45545.020

## Selective disruption of Shh signaling in astrocytes produces mild reactive gliosis

We previously demonstrated that cortical astrocytes in *Gfap Smo* CKO mice upregulate GFAP expression and exhibit cellular hypertrophy, two classic hallmarks of reactive astrogliosis (*Garcia et al., 2010*). Astrocytes exhibiting these features were broadly distributed across cortical layers (*Figure 6*), in contrast to the distribution of Gli1 astrocytes which are found predominantly in layers IV and V. In addition to upregulation of GFAP, astrocytes in *Gfap Smo* CKO mice show dramatic changes in morphological structure. Sholl analysis revealed no difference in the number of primary branches in *Gfap Smo* CKO mice compared to WT controls (7.2 and 7.5 branches, WT [*n* = 9 cells] and *Gfap Smo* CKO [*n* = 9 cells], respectively, three animals per genotype). However, astrocytes from *Gfap Smo* CKO animals did show an increase in the number of higher order branches, the length of the longest tree (136.6 ± 17 and 212.1 ± 28 µm for WT and *Gfap Smo* CKO, respectively, p=0.035), and the total length of all processes (397 ± 66 and 674 ± 91 µm for WT and *Gfap Smo* CKO, respectively, p=0.025) compared to WT controls (*Figure 6*). Consistent with this, there was a significant increase in the number of branches intersecting concentric shells at various distances from the soma (*Figure 6*). Notably, in *CK2 Smo* CKO mice, GFAP staining in the cortex was indistinguishable from WT controls (*Figure 6—figure supplement 1*), suggesting that astrocytes exhibit dramatic changes in morphological structure following astrocytic, but not neuronal, disruption of Shh signaling. Such changes in morphology are consistent with mild reactive gliosis, a

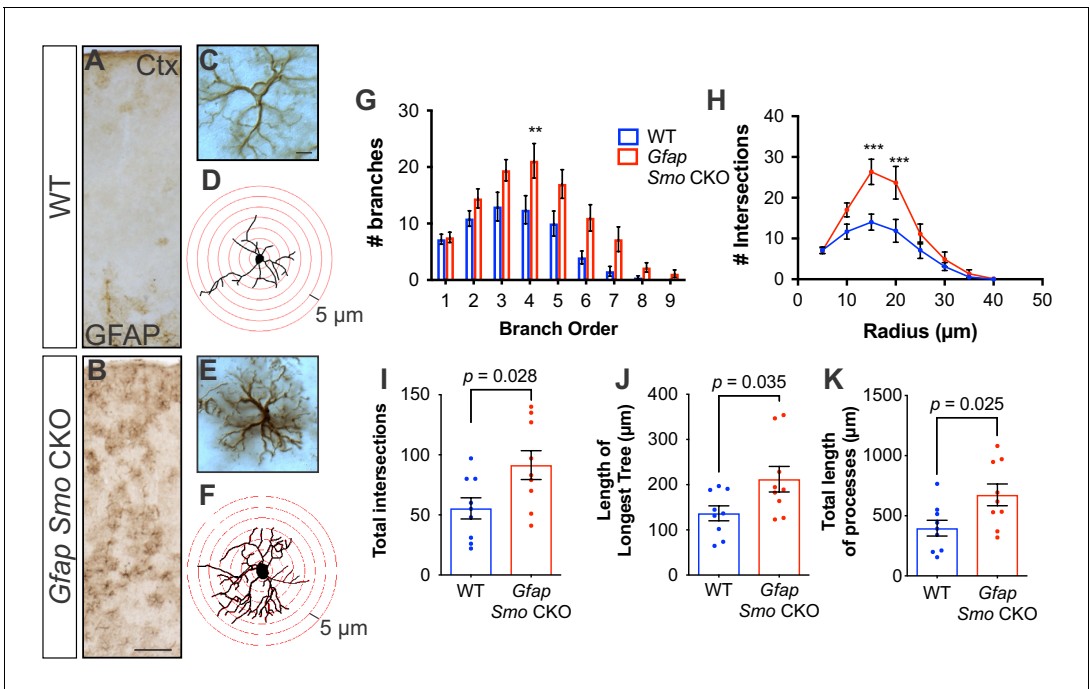

**Figure 6.** Disruption of Shh signaling in astrocytes results in widespread reactive changes in morphology. (A–B) Brightfield immunohistochemistry reveals a reactive upregulation of GFAP across cortical layers in *Gfap Smo* CKO astrocytes. Scale bar, 125 µm. (C and E) Representative high-magnification images of both WT (C) and *Gfap Smo* CKO (E) astrocytes are shown. Scale bar, 10 µm. (D and F) Representative traces of GFAP expression from WT (D) and *Gfap Smo* CKO (F) cortical astrocytes. (G and H) Sholl analysis shows significant increases in complexity of *Gfap Smo* CKO astrocytes compared to WT controls. Statistical significance was assessed by one-way ANOVA with Bonferroni's for multiple comparisons. (I–K) Quantification of various morphological features of traced astrocytes. Statistical analysis performed by one-way ANOVA (G and H) and unpaired Student's t-test (I–K). Data points represent individual cells. Graphs represent mean value ± SEM. *n* = 3 animals per genotype.

DOI: https://doi.org/10.7554/eLife.45545.023

The following source data and figure supplement are available for figure 6:

**Source data 1.** Sholl analysis of cortical astrocytes.
DOI: https://doi.org/10.7554/eLife.45545.025

**Figure supplement 1.** Reactive gliosis does not occur when neuronal Shh signaling is disrupted.
DOI: https://doi.org/10.7554/eLife.45545.024

complex cellular response of astrocytes to disturbances in physiological homeostasis (*Sofroniew, 2015*). Notably, the distribution of reactive astrocytes in *Gfap Smo* CKO mice is broader than that defined by Gli1 expression, arguing against the idea that Shh signaling regulates the intrinsic state of astrocytes. Rather, reactive gliosis may instead reflect a cellular response to the mild, but persistent, aberrations in neuronal activity observed in *Gfap Smo* CKO mice.

## Discussion

The dynamic processes underlying organization of synapses during postnatal development are essential for establishing functional neural circuits. In this study, we demonstrate that astrocytes modulate refinement and reorganization of synaptic connectivity in the postnatal cortex in a Shh-dependent manner. Our data show that Gli1 astrocytes are enriched in deep cortical layers with a relative paucity in upper cortical layers. Selective disruption of Shh signaling in astrocytes leads to an overabundance of spines on the apical dendrites of layer V, but not layer II/III, cortical neurons. The overabundance of spines emerges during postnatal development, continues into adulthood and is accompanied by a reduction in dynamic turnover. *Gfap Smo* CKO mice exhibit a pronounced reduction in Kir4.1 as well as an increase in neuronal excitability. Finally, we show that cortical astrocytes exhibit dramatic changes in morphology and GFAP expression, a phenotype consistent with mild reactive gliosis. Our data identify an essential role for astrocyte modulation of neuronal activity during postnatal development, facilitating activity-dependent reorganization of synaptic connectivity.

To accomplish selective disruption of Shh signaling in astrocytes, we used a Cre driver in which expression is regulated by the full-length mouse Gfap gene (*Garcia et al., 2004*). Single cell analysis in *Gfap-Cre*;Ai14 mice showed that nearly all cortical astrocytes undergo Cre-mediated recombination, while the fraction of recombined cortical neurons is less than 2%. Importantly, we observed recombined neurons predominantly in superficial layers, consistent with early Cre activity during late embryogenesis when layer II/III neurons are being generated. Notably, despite a small population of *Smo* null neurons in layer II/III, changes in spine density are not observed in these cells. In addition, a more complete deletion of *Smo* in excitatory neurons using the *CamKIIαCre* driver fails to alter spine density in layer V neurons, arguing against non-specific effects arising from the small population of recombined neurons in *Gfap Smo* CKO mice. Importantly, Gli1 expression is nearly completely lost in *Gfap Smo* CKO mice, with no loss in the number of total astrocytes (*Garcia et al., 2010*), demonstrating effective disruption of Shh activity in these mice. Because a small number of recombined oligodendrocytes were observed in *GfapCre*;Ai14 mice, the possibility that these cells may contribute to some of the observed phenotypes cannot be ruled out. However Gli1 expression is restricted predominantly to astrocytes in the adult forebrain (*Garcia et al., 2010*). Thus, any contribution from other cell types must be mediated by non-canonical, Gli-independent mechanisms.

In the postnatal cortex, Shh signaling from layer V neurons mediates synaptic connectivity with their layer II/III presynaptic partners (*Harwell et al., 2012*). Here, we show that, in addition to neuronal Shh signaling, Shh activity in astrocytes is required for the establishment and maintenance of cortical circuits. This is supported by two key observations in our study. First, we find that in *Gfap Smo* CKO mice, apical dendrites of cortical neurons exhibit an increase in spine density that emerges during postnatal development and persists into adulthood. Importantly, genetic deletion of *Smo* in mature neurons instead of astrocytes failed to show any differences in spine density, pointing to a specific role for Shh signaling in astrocyte regulation of synaptic connectivity. Interestingly, deletion of Shh ligand using an EmxCre driver produces a reduction in spine density of basal dendrites (*Harwell et al., 2012*), whereas we found no difference in basal dendrite spine density, suggesting that Shh acts in distinct, cell type-specific ways to regulate synaptic connectivity of local cortical circuits. Indeed, whereas astrocytes express Gli1, neurons do not (*Garcia et al., 2010*), indicating differential transduction of Shh through canonical and non-canonical, Gli-independent pathways, respectively, in different cell types. Second, our data show fewer filopodia in juvenile *Gfap Smo* CKO mice compared to WT controls along with a concomitant increase in spine stability. In the adult, spines continue to show deficits in dynamic turnover, demonstrating long-term impairments in structural plasticity. This suggests that Shh signaling is required not only for identifying appropriate synaptic partners during development, but also plays an important role in mediating synaptic plasticity.

In the postnatal and adult brain, we and others identified neurons as the source of Shh (*Harwell et al., 2012*; *Garcia et al., 2010*; *Farmer et al., 2016*; *Álvarez-Buylla and Ihrie, 2014*).

However non-neuronal sources, including astrocytes have been identified in the injured brain (*Amankulor et al., 2009*; *Sirko et al., 2013*). Beyond the CNS, epithelial cells have been reported as a source of Shh in the developing dentate gyrus (*Choe et al., 2015*). It will be interesting to examine whether other non-neuronal sources are available to trigger astrocyte transduction of Shh.

We previously demonstrated that Gli1 expression is restricted to a subpopulation of cortical astrocytes (*Garcia et al., 2010*). Here, we extend these findings and demonstrate that the distribution of Gli1 astrocytes in the cortex is not uniform across cortical layers, but rather, occurs in a laminar-specific fashion. Cortical layers have been historically defined by neuronal populations and their specific functional properties and connectivity. However, emerging evidence suggests that astrocytes also exhibit cortical lamination patterns based on distinct gene expression profiles (*Bayraktar, 2018*; *Miller et al., 2019*). Our data show that Gli1 astrocytes are enriched in layers IV and V with a relatively sparse distribution in layer II/III. Interestingly, a recent study reported a population of astrocytes with a similar distribution that show enrichment of the Shh pathway (*Miller et al., 2019*). These cells were also shown to regulate spine density of Layer V cortical neurons. It will be interesting to examine whether these cells correspond to Gli1 astrocytes reported in this study. Whereas astrocytes transducing Shh are enriched in deep cortical layers, the BMP antagonist, chordin-like 1 (Chrdl1) is preferentially enriched in upper layer cortical astrocytes (*Blanco-Suarez et al., 2018*), suggesting that Gli1 and Chrdl1 may identify two distinct, but regionally complementary, astrocyte populations. Alternatively, Shh signaling may actively repress Chrdl1 expression in deep layer astrocytes. Notably, our observation that Gli1 astrocytes only comprise a fraction of astrocytes in layers IV and V suggests that, in addition to the laminar-specific distribution of these cells, there may be additional heterogeneity of astrocytes even within a given cortical layer. Whether and how the laminar distribution of astrocytes with distinct gene expression profiles confers functional specialization is not well understood and requires further study.

Interestingly, we observed that the disturbances in synaptic organization of cortical neurons in *Gfap Smo* CKO mice occurs selectively in apical dendrites of layer V, but not layer II/III, neurons. Since apical dendrites of layer V neurons traverse through layer II/III, this suggests that Shh-dependent regulation of synaptic refinement is not mediated by direct interaction between Gli1 astrocytes and synapses. One possibility is that Gli1 astrocytes interact with neuronal soma in layer V and effectively modulate their excitability. Our observation that Kir4.1 expression in cortical astrocytes is regulated by Shh signaling suggests that *Gfap Smo* CKO mice may experience deficits in local buffering of extracellular $K^+$ leading to perturbations in neuronal activity. Indeed, a recent study demonstrated that Kir4.1 surrounds neuronal soma in the lateral habenula and regulates their firing properties (*Cui et al., 2018*). Astrocytes play essential roles in buffering extracellular $K^+$ through Kir4.1, and several studies demonstrate that loss of function of Kir4.1 impairs $K^+$ uptake and increases neuronal excitability (*Djukic et al., 2007*; *Tong et al., 2014*; *Sibille et al., 2014*). Alternatively, increased neuronal excitability may be due to disruptions in glutamate clearance. Although we did not observe changes in expression of either *Glt1* or *Glast*, deficits in Shh-dependent trafficking, localization or transporter function would lead to impairments in astrocyte-mediated glutamate clearance. Further work is needed to definitively identify the precise mechanism underlying the abnormal firing properties of neurons in *Gfap Smo* CKO mice.

The establishment of appropriate synapse number and connectivity are tightly regulated processes that are profoundly shaped by experience or activity (*Zuo et al., 2005b*; *Yang et al., 2009*). Our observation that *Gfap Smo* CKO mice show increases in spontaneous neuronal activity and excitability, accompanied by a pronounced increase in spine density suggests that astrocytes contribute to activity-dependent processes that shape the organization of developing neural circuits in a Shh-dependent manner. The deficits in synaptic organization observed in *Gfap Smo* CKO mice persist for the life of the animal. These deficits are not limited to synapse number but also include lower rates of spine turnover, suggesting long-lasting deficits in structural plasticity. Together with the observation that Shh guides local synaptic connectivity of cortical neurons (*Harwell et al., 2012*), these studies reveal that Shh signaling exerts considerable influence on the establishment and maintenance of cortical circuits.

# Materials and methods

## Key resources table

| Reagent type (species) or resource | Designation | Source or reference | Identifiers | Additional information |
|---|---|---|---|---|
| Genetic reagent (*M. musculus*) | *Gli1*$^{CreER/+}$ | The Jackson Laboratory; *Ahn and Joyner, 2005* | stock no. 7913 | from Alexandra Joyner, MSKCC |
| Genetic reagent (*M. musculus*) | Ai14 (B6.Cg-Gt(ROSA)26Sor$^{tm14(CAG-tdTomato)Hze}$/J) | The Jackson Laboratory; *Madisen et al., 2010* | stock no. 7914 | |
| Genetic reagent (*M. musculus*) | *Gfap-Cre*, line 73.12 | The Jackson Laboratory; *Garcia et al., 2004* | stock no. 12886 | from Michael Sofroniew, UCLA |
| Genetic reagent (*M. musculus*) | *Smo*$^{fl/fl}$ | The Jackson Laboratory; *Long et al., 2001* | stock no. 4526 | |
| Genetic reagent (*M. musculus*) | *CamKinaseIIα-Cre (CamKIIα-Cre)* | The Jackson Laboratory; *Tsien et al., 1996* | stock no. 5359 | |
| Genetic reagent (*M. musculus*) | *Thy1*-GFPm | The Jackson Laboratory; *Feng et al., 2000* | stock no. 7788 | |
| Chemical compound, drug | tamoxifen | Sigma | catalog no. T5648 | |
| Antibody | Polyclonal rabbit anti-GFP | Millipore | catalog no. AB3080 | 1:5000 (brightfield); 1:1000 (fluorescence) |
| Antibody | Polyclonal rabbit anti-Kir4.1 | Millipore | catalog no. AB5818 | 1:1000 (fluorescence) |
| Antibody | Polyclonal rabbit anti-Gfap | Agilent Dako | catalog no. Z033429-2; | 1:20,000 (brightfield) |
| Antibody | Monoclonal mouse anti-NeuN | Millipore | catalog no. MAB377 | 1:1000 (fluorescence) |
| Antibody | Polyclonal rabbit anti-RFP | MBL International | catalog no. PM005 | 1:500 (fluorescence) |
| Antibody | Monoclonal rat anti-Ctip2 | Abcam | catalog no. ab18465 | 1:2000 (fluroescence0 |
| Antibody | Polyclonal sheep anti-CAII | Bio-Rad | catalog no. AHP206 | 1:2000 (fluorescence) |
| Antibody | Polyclonal rabbit anti-S100B | Agilent Dako | catalog no. Z0311 | 1:1000 (fluorescence) |
| Antibody | Polyclonal rabbit anti-Olig2 | Millipore | catalog no. AB9610 | 1:1000 (fluorescence) |

*Continued on next page*

*Continued*

| Reagent type (species) or resource | Designation | Source or reference | Identifiers | Additional information |
|---|---|---|---|---|
| Software, algorithm | computer assisted tracing of dendrites and spines; Sholl analysis of astrocytes | Neurolucida, MBF Biociences | | Williston, VT |
| Software, algorithm | MATLAB-based custom-written code for longitudinal analysis of spine turnover | This paper | | *Figure 3—source code 1* |

## Animals

All experiments were approved by Drexel University's Institutional Animal Care and Use Committee and were conducted according to approved protocols. We used the following strains of transgenic mice on the C57BL/6 background: *Gli1CreER/+* (*Ahn and Joyner, 2005*), Ai14 (*Madisen et al., 2010*), *Smofl/fl* (*Long et al., 2001*), *Gfap-Cre* (*Garcia et al., 2004*), *CaMKIIa-Cre* (*Tsien et al., 1996*) and *Thy1-GFPm* (*Feng et al., 2000*). Mice of either sex were included in these studies.

## Tamoxifen

Tamoxifen was administered as previously described (*Garcia et al., 2010*; *Allahyari et al., 2019*). Briefly, tamoxifen (Sigma, T5648) was dissolved in corn oil to a final concentration of 20 mg/ml. Adult *Gli1CreER/+*;Ai14 mice received 250 mg/kg tamoxifen by oral gavage for three consecutive days, and tissues were analyzed two – three weeks later.

## Immunohistochemistry

Mice were deeply anesthetized by intraperitoneal injection of ketamine/xylazine/acepromazine. Animals were then transcardially injected with 100 units heparin and perfused with phosphate-buffered saline (PBS) followed by 4% paraformaldehyde (PFA, Sigma-Aldrich, St. Louis, MO). Brains were fixed overnight at 4℃ and then transferred to 30% sucrose for at least 48 hr or until sectioned. Brains were cryosectioned (Leica CM3050S, Wetzlar, Germany) and collected in 40 μm sections. Sections were stored in 0.1M Tris-buffered saline (TBS) with 0.05% sodium azide at 4℃. Immunohistochemistry was performed using the following primary antibodies: rabbit anti-GFP (AB3080; Millipore, Burlington, MA), rabbit anti-Kir4.1 (AB5818; Millipore), rabbit anti-GFAP (Z033429-2; Agilent DAKO, Santa Clara, CA), mouse anti-NeuN (MAB377; Millipore), rabbit anti-RFP (PM005; MBL International, Woburn, MA), rat anti-Ctip2 (ab18465; Abcam, Cambridge, UK), sheep anti-CAII (AHP206; Bio-Rad), rabbit anti-S100β (Z0311, Agilent), and rabbit anti-Olig2 (AB9610; Millipore). Sections were washed three times in 0.1M TBS at room temperature (10 min/wash). Sections were then blocked in 10% Normal Serum with 0.5% Triton-X (Sigma-Aldrich) for one hour at room temperature and incubated with primary antibody in 0.5% Triton-X at 4℃ overnight. For fluorescent immunohistochemistry, sections were rinsed the following day in 0.1M TBS and incubated in Alexafluor-conjugated secondary antibodies with 10% Normal Serum and 0.1M TBS and incubated at room temperature for two hours. The sections were then rinsed in 0.1 TBS and incubated in DAPI (1:50,000; Life Technologies, Carlsbad, CA) for 15 min. Sections were rinsed again in 0.1M TBS and mounted onto microscope slides (Fisherbrand, Waltham, MA) and coverslipped using ProLong Gold Antifade Mountant (Invitrogen, Carlsbad, CA) and Fisherfinest Premium Cover Glass. For brightfield immunohistochemistry, sections were rinsed as previously described but were incubated with biotinylated secondary antibodies against the primary antibody species (Vector Laboratories, Burlingame, CA) for one hour. Sections were then placed in Avitin-Biotin Complex solution (Vector) for one hour and visualized using 3,3'-diaminobenzidine (DAB Peroxidase Substrate Kit, Vector). Sections were then mounted onto slides and coverslipped with DPX Mountant (Fisher).

## Quantification of dendritic spine density

Analysis of spine density was performed in a blinded study design. WT controls were derived from Cre-negative littermates. Neurons were traced using Neurolucida (MicroBrightField Biosciences, Williston, VT) and individual spines were marked. Apical dendrites of layer V neurons were counted from ~100 μm below the primary bifurcation through the apical tuft. Apical dendrites of layer II/III cells were analyzed just below the primary bifurcation through the apical tuft. Basal dendrites were analyzed beginning ~50 μm from the soma through the end of the processes. CA1 hippocampal neurons were analyzed from just below the primary bifurcation through the apical tuft. All analysis was performed on an upright Zeiss microscope using a 63X oil objective.

## Cranial window

Cranial window surgeries were adapted from a published protocol (*Holtmaat et al., 2009*). Immediately prior to each surgery, mice received a subcutaneous injection of carprofen (Rimadyl, 5 mg/kg), and a follow-up injection was given 24 hr after surgery. Mice were deeply anesthetized with isoflurane (5%) in an induction chamber, and then transferred to a stereotaxic apparatus where they inhaled 1.5% isoflurane continuously for the duration of the surgery. The scalp was then removed and a metal bar (~1 cm long) with threadings for screws was affixed to the skull, using superglue and dental acrylic to form a head post that sealed off the skin while leaving the right parietal bone exposed. After allowing 24 hr for the glue and acrylic to set, the mouse was returned to the stereotaxic apparatus and head-fixed using the headpost. A circular craniotomy ~3 mm in diameter and centered 2.5 mm lateral and posterior to bregma was performed. The exposed dura was treated with saline-soaked gel foam until any minor bleeding ceased. Finally, a 3 mm glass coverslip was gently pressed onto the dura and sealed in place using superglue and dental acrylic. Adult mice were allowed at least 1 week for recovery. To mitigate the tendency of rapid bone growth to destabilize the windows at younger ages, juvenile mice were allowed 24 hr to recover.

## Two-photon imaging

A commercial two-photon microscope (Bruker, Billerica, MA) with a tunable Ti:Sapphire laser (Coherent, Santa Clara, CA) was used for all experiments. The laser was set to 920 nm and power was tuned on the sample as necessary to obtain consistently high-quality images without damaging the tissue. Mice were head-fixed and anesthetized with continuous isoflurane (1–1.5%) during imaging. Under a 20X water immersion objective, GFP-expressing neurons were identified visually and then imaged down to the cell body to determine layer. From the apical dendrite tuft, individual dendrite segments that projected primarily along the xy-plane were selected for analysis, and z-stacks (0.5 μm step size) were collected. To track individual dendrites, repeated imaging was generally performed on the following schedule (in days, relative to first imaging session): 0, 1, 2, 7, 14, 21, 28, 42. Some data from studies using other schedules were also included. For juvenile mice, this schedule was: 0, 1, 2.

## Analysis of spine dynamics

Every effort was made to use Cre-negative, littermate controls for these experiments. However, it was not always possible to match littermate controls in the final data sets presented here, either due to the absence of the GFP allele in the Cre-negative controls, or due to the loss of head caps securing the cranial windows, over time. However, it should be noted that all data presented here represent Cre-negative controls from the appropriate strain, if not necessarily from the same litter.

To analyze spine turnover, we performed side-by-side manual comparison of dendritic protrusions using ImageJ. Z-stacks from two time points were analyzed, and individual protrusions were identified as present or absent in each stack. For each mouse, a minimum of 150 protrusions were analyzed, although the average number of protrusions analyzed per mouse was 250–300. These comparisons were conducted by trained observers blinded to mouse genotype. Long, thin protrusions lacking a bulbous head were identified as filopodia, and any other prominent dendritic protrusion that extended >0.4 μm from the shaft was counted as a spine. The filopodial fraction was calculated as the proportion of all analyzed features present at the first imaging timepoint that were identified as filopodia. The position and dynamic status (stable/eliminated/formed) of each spine was recorded using ImageJ's CellCounter plugin. Each neuron's turnover ratio was calculated as:

$$TO = (N_{elim} + N_{form})/(N_{elim} + N_{form} + 2N_{stable})$$

For longitudinal analysis of spine lifetime, the individual features identified in our manual ImageJ analysis were then imported into custom written MATLAB (Mathworks, Natick, MA) code and manually tracked across all imaged time-points using custom-written software. This code facilitated the efficient manual tracking of each feature by iteratively estimating the z-plane, within each stack, in which that feature would be expected to appear, assuming a linear relationship. Using a graphical user interface, the user was presented with a series of frames centered on that plane and allowed to make a determination of the feature's presence or absence simply by left-clicking on the feature, or in the case of absence right-clicking on the location it otherwise would have been, using the best frame presented (*Figure 3—figure supplement 2*). Due to the iterative nature of this procedure, each dataset was run through at least twice to ensure that not only were the determinations of presence/absence accurate, but that the marked locations of every feature were optimal.

Spines that were observed consistently across all time points were classified as stable. Spines that disappeared at some time point and were observed again in a subsequent time point were classified as recurrent. Spines that disappeared and never reappeared were classified as transient. To compare long-term dynamics between WT and *Gfap Smo* CKO mice, we examined the survival curves, $S(t)$, for each mouse, where:

$$S(t) = \# \,of\, spines\, consistently\, present\, to\, time\, point \,/\, \# \,of\, spines\, initially\, observed$$

Survival curves for each mouse were pooled by genotype and fit to a single-phase exponential decay model:

$$S(t) = S_p + S_i{}^* \exp(-t/\tau)$$

where $S_p$ and $S_i$ represent the relative fractions of permanent and impermanent spines respectively and τ is the characteristic lifespan of impermanent spines. Fitting and statistical comparison of survival data was performed with PRISM 6 (GraphPad, San Diego, CA) software.

To analyze morphology, we examined data from the first imaged day, and classified individual protrusions as filopodia, intermediate, or mushroom spines. For each protrusion, a mean projection image was generated, averaging the frame on which the protrusion was manually marked in the original analysis and the two neighboring frames (1 above and one below, effectively averaging over 1.5 um of depth in the z axis). Each image was cropped, centering around the protrusion's XY coordinates and a marker was drawn on the image to indicate which protrusion was under consideration. These images were then presented to the scorer sequentially for scoring. Protrusions exhibiting a bright, bulbous head were scored as mushroom. Thin protrusions exhibiting a relatively dim, small head, or no head, were scored as intermediate. Dim, elongated protrusions that lacked any head were scored as filopodia.

## Transcript quantification

Mice were deeply anesthetized by intraperitoneal injection of ketamine/xylazine/acepromezine before rapid decapitation. Cortices were dissected into TRIzol reagent (Thermo-Fisher, Waltham, MA) and RNA was extracted according to the standard protocol. RNA was then purified with an RNeasy Micro Kit (Qiagen, Hilden, Germany) and subsequently reverse transcribed to cDNA using the High-Capacity cDNA Reverse Transcription Kit (Thermo-Fisher). Droplet digital PCR (ddPCR) was performed with the QX200 Droplet Digital PCR System using Evagreen Supermix (Bio-Rad, Hercules, CA). Alternatively, for *Megf10* and *Mertk*, qPCR was performed with the CFX96 Touch Real-Time PCR Detection System (Bio-Rad) using PowerUp SYBR Green Master Mix (Thermo-Fisher). PrimePCR ddPCR Expression EvaGreen Assay (Bio-Rad) was used for *Gapdh* primers; all other primers were designed using NCBI Primer-BLAST as follows: *Kcnj10* F, GCTGCCCCGCGATTTATCAG; *Kcnj10* R, AGCGACCGACGTCATCTTGG; *Glast* F, TCCTCTACTTCCTGGTAACCC; *Glast* R, TCCACACCATTG TTCTCTTCC; *Glt1* F, CATCAACAGAGGGTGCCAAC; *Glt1* R, CACACTGCTCCCAGGATGAC; *Megf10* F, CTCACTGCTCTGTCACTGGGTG; *Megf10* R, GGTAGCTGATTCTGTGCCGTGT; *Mertk* F, AAACTGCATGTTGCGGGATGAC; *Mertk* R, TCCCACATGGTCACGCCAAA. ddPCR absolute quantification of targets was calculated using Quantasoft Version 1.7 software (Bio-Rad). Samples were

run in triplicate, and *Gapdh* was quantified for each sample as a loading control with no difference detected between sample groups.

## Slice electrophysiology

Mice aged P21 were anesthetized with intraperitoneal injection of euthasol (0.2 ml/kg) and brains were rapidly removed. 300 µm coronal slices were cut using a vibratome tissue slicer (Leica) and transferred to a holding chamber, submerged in oxygenated artificial cerebrospinal fluid (ACSF, in mM: 124 NaCl, 2.5 KCl, 1.25 $NaH_2PO_4$, 2 $CaCl_2$, 1 $MgSO_4$, 26 $NaHCO_3$, and 10 dextrose, pH 7.4) at 36˚C for 1 hr and then maintained at room temperature. Individual slices containing the barrel cortex were placed into a recording chamber immersed in oxygenated ACSF mounted on an Olympus upright microscope (BX51). Neurons were visualized with infrared differential interference video microscopy.

Whole-cell current clamp was used to record action potentials from layer V pyramidal cells using patch electrodes with an open tip resistance of 7–10 MΩ. Patch electrodes were filled with potassium gluconate internal solution (in mM): 120 potassium gluconate, 20 KCl, 4 ATP-Na, 0.3 $Na_2GTP$, 5 Na-phosphocreatine, 0.1 EGTA, 10 HEPES, pH 7.3, 305 mOsmol/l) and action potential responses were measured in response to various step currents from −300 pA to +650 pA with 50 pA increments. Action potentials were described as spike numbers per depolarized current injections. The resting membrane potential, input resistance, action potential (AP) threshold, AP half-width, and peak AP amplitude were also measured.

Whole-cell voltage-clamp recordings were obtained from layer V pyramidal cells using patch electrodes with $Cs^+$-containing solution (in mM): 120 Cs-gluconate, five lidocaine, 6 $CaCl_2$, 1 $Na_2ATP$, 0.3 $Na_2GTP$, and 10 HEPES (pH 7.3 adjusted by CsOH). To record spontaneous excitatory postsynaptic currents (sEPSCs), cells were held at a membrane potential of −70 mV in the presence of $GABA_A$ receptor antagonist picrotoxin (PTX; 100 µM, Sigma-Aldrich) and recorded for 5 min. Miniature EPSCs (mEPSCs) were then recorded for an additional 5 min in the presence of both picrotoxin and tetrodotoxin (TTX, 0.5 µM, Hello Bio, Princeton, NJ). All recordings were conducted with Axon MultiClamp 700B amplifier (Molecular Devices, San Jose, CA). The sEPSCs and mEPSCs frequency and amplitudes were measured by averaging five sweeps from the onset of recording with Clampfit 9.2 software (Molecular Devices). The s/mEPSCs were detected and characterized using a sample template for the 5 min data period. Based upon the selected sample template, the frequency (number of event detections) and amplitude of events were measured using a threshold set in Clampfit.

## Statistical analyses

Statistical analyses used for various datasets are indicated both in the figure legends and in the text. Prism six software (GraphPad) was used for all analyses and to generate graphs.

## Acknowledgements

We would like to thank Pooja Sakthivel and Nikhil Karmacharya for technical assistance and Dr. Monica Truelove-Hill for assistance with statistical analyses. We also thank Dr. Michael Akins for helpful discussions.

## Additional information

### Funding

| Funder | Grant reference number | Author |
|---|---|---|
| National Institute of Neurological Disorders and Stroke | 1R01NS096100 | A Denise R Garcia |
| National Institute of Mental Health | 5R21MH110724 | A Denise R Garcia |
| Pennsylvania Department of Health | CURE | Wen-Jun Gao<br>A Denise R Garcia |
| National Institute of Neurological Disorders and Stroke | F99NS105185 | Austin A Coley |

| National Institute of Mental Health | R01MH085666 | Wen-Jun Gao |
|---|---|---|
| National Institute of Neurological Disorders and Stroke | K01NS089720 | Corey C Harwell |
| National Institute of Neurological Disorders and Stroke | R01NS102228 | Corey C Harwell |
| Genise Goldenson | Junior Faculty Award | Corey C Harwell |
| Alice and Joseph Brooks Fund | Postdoctoral Fellowship | Yajun Xie |
| Louis Perry Jones | Postdoctoral Fellowship | Yajun Xie |
| National Institute of Mental Health | 7K01MH097957 | A Denise R Garcia |

The funders had no role in study design, data collection and interpretation, or the decision to submit the work for publication.

### Author contributions

Steven A Hill, Andrew S Blaeser, Investigation, Writing—original draft, Writing—review and editing; Austin A Coley, Yajun Xie, Formal analysis, Investigation; Katherine A Shepard, Formal analysis, Investigation, Writing—review and editing; Corey C Harwell, Wen-Jun Gao, Writing—review and editing, Analysis and interpretation of data; A Denise R Garcia, Conceptualization, Resources, Supervision, Funding acquisition, Writing—original draft, Writing—review and editing

### Author ORCIDs

A Denise R Garcia (iD) https://orcid.org/0000-0001-5809-3543

### Ethics
Animal experimentation: This study was performed in strict accordance with the recommendations in the Guide for the Care and Use of Laboratory Animals of the National Institutes of Health. All of the animals were handled according to approved institutional animal care and use committee (IACUC) protocols (#20476) of Drexel University. All surgery was performed under isoflurane or ketamine/xylazine anesthesia, and every effort was made to minimize suffering.

### Decision letter and Author response
Decision letter https://doi.org/10.7554/eLife.45545.028
Author response https://doi.org/10.7554/eLife.45545.029

# Additional files

### Supplementary files
• Transparent reporting form
DOI: https://doi.org/10.7554/eLife.45545.026

### Data availability
All data generated or analysed during this study are included in the manuscript and supporting files. Source data files have been provided for Figures 1-6.

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
