## [Decision Letter]

Thank you for submitting your article "Sonic hedgehog signaling in astrocytes mediates cell-type-specific synaptic organization" for consideration by *eLife*. Your article has been reviewed by two peer reviewers, and the evaluation has been overseen by a Reviewing Editor and Didier Stainier as the Senior Editor. The reviewers have opted to remain anonymous.

The reviewers have discussed the reviews with one another and the Reviewing Editor has drafted this decision to help you prepare a revised submission.

Summary:

This study examined the role of Sonic hedgehog (Shh) signaling in shaping dendritic spine development, excitatory synapse formation and neuronal excitability in the mouse cerebral cortex through modulation of astrocytes. Prior studies have shown that neuronal release of Shh controls Kir4.1 expression in Bergmann glia and some astrocytes, and selective loss of Smoothened (Smo), an obligate Shh co-receptor, leads to astrocyte hypertrophy and upregulation of GFAP. Although neuronal Shh is involved in synapse formation between Layer 5 and Layer 2/3 cortical neurons, functional links between astrocyte Shh signaling and neuronal development have not been demonstrated. Here, the authors report that Gli1, the transcriptional effector of Shh signaling, is expressed in astrocytes in a laminar-specific manner, with significant expression in astrocytes within cortical Layers 4 and 5. Conditional deletion of Smo from astrocytes led to accelerated development of spines on the apical dendrites of Layer 5 pyramidal neurons and greater spine density in adulthood. Using in vivo time lapse imaging, the authors found that dendritic spines from juvenile astrocytic Smo-cKO mice had fewer filopodia, and adult cKO mice had a larger proportion of stable spines. Furthermore, whole-cell patch clamp recording of Layer 5 neurons showed that both spontaneous and miniature EPSCs were enhanced (frequency and amplitude) when astrocyte Smo was deleted. Astrocytes with Smo deletion exhibited lower Kir4.1 immunoreactivity, upregulation of GFAP and enhanced process arborization, changes that were accompanied by an increase in excitability of layer 5 neurons, suggesting that Shh signaling in astrocytes leads to both structural and functional changes in nearby neurons. Together, these studies substantially extend our knowledge about the molecular mechanisms used to coordinate neuron-astrocyte interactions during development and point to an important role for Shh in shaping circuit formation and neuronal development by modulating the properties of astrocytes.

Essential revisions:

1) The authors need to show some form of validation that mGFAP-Cre is efficiently reducing Smo expression in astrocytes. It is possible, even with constitutive Cre expression, that recombination may not occur at both floxed alleles.

2) In light of the Harwell et al., 2012 study which showed that pyramidal neurons utilize Shh signaling to regulate dendrite growth and spines in Layer V neurons, and the fact that Smo deletion with mGFAP-Cre is not exclusive to astrocytes, the authors should consider softening their conclusions. Given the limited recombination in other cell types observed with the mGFAP-Cre line, it is likely that the effects observed are related to loss of Smo in astrocytes. However, it is still possible that loss of Smo in other cells contributes to the observed changes. Also, the authors should soften their conclusion that Shh from neurons is mediating the observed changes given that this aspect was not experimentally tested. Regardless of where Shh comes from, it would not alter the conclusion that Shh-dependent signaling in astrocytes alters dendritic spine dynamics and neuronal excitability.

3) The authors should improve illustration of the spine development changes. It is difficult to determine what are spines in Figure 2A and B. This evidence needs to be substantially improved and they should provide representative examples for all of the time points and all of the areas sampled in this figure. The cKO dendrite in Figure 2 seems to be substantially thicker than the WT control. Is this specific to this image, or is this consistent across neurons? If this is a phenotype, please quantify. If not, please choose a different representative dendrite. In Figure 3, it is unclear whether n should refer to spines or to animals. As animals have been subjected to different surgeries, one could argue that n should refer to animals. Please provide further justification for the determination of n in these experiments. The authors should provide additional details about how spines were analyzed and the effectiveness of the custom software in tracking spines should be demonstrated in supplementary material.

4) The results indicate that there are fewer filopodia in juvenile Smo-cKO animals. The authors also cite references supporting a strong correlation between dendritic spine morphology and maturity. To establish whether there are simply fewer filopodial extensions in the Smo-cKO animals, or whether the dendritic spines themselves are more mature, the authors should classify spine morphology as filopodia/intermediate/mushroom-shaped.

5) The authors provide evidence that Shh signaling in astrocytes regulates neuronal excitability by controlling Kir4.1 expression in astrocytes. The data presented only show a correlation between astrocytes changes (decrease in Kir4.1 expression, increase in GFAP and hypertrophy) and neuronal hyperexcitability. The studies would be more impactful if the authors were to explore whether neuronal excitability was directly related to Kir4.1 downregulation, by assessing whether block of Kir4.1 by Ba^+^ or desipramine in WT neurons elicits a similar increase in excitability.

---

## [Author Response]

Essential revisions:1) The authors need to show some form of validation that mGFAP-Cre is efficiently reducing Smo expression in astrocytes. It is possible, even with constitutive Cre expression, that recombination may not occur at both floxed alleles.

This comment raises two distinct issues – efficiency of recombination and specificity. We have performed qPCR for Smo on whole cortex from mGFAP Smo CKO mice and wild type littermate controls. We observed a 50% reduction in gene expression in CKO mice. The remaining 50% is likely from neurons, which are known to use Boc-mediated Shh signaling (Harwell et al., 2012). These data are now included in Figure 2—figure supplement 1. Indeed, we cannot rule out the possibility that recombination may not occur at both floxed alleles. However we previously demonstrated that in mGFAP Smo CKO mice possessing a *Gli1^lacZ/+^* allele, there is a nearly complete loss of Gli1 expression, with no change in the number of astrocytes (Garcia et al., 2010), demonstrating effective disruption of Shh signaling, and arguing against inefficient recombination. Nevertheless, even if both alleles fail to undergo recombination, this does not change our underlying observation that targeted disruption of Shh signaling in astrocytes produces deficits in establishing the mature circuitry of the cortex. Indeed, it should be noted that we observe reliable structural, physiological, and molecular phenotypes, suggesting that if recombination is inefficient in our mice, even more severe or additional phenotypes, would be expected.

Specificity of recombination in astrocytes is demonstrated by our single cell analysis in mGFAPCre;Ai14 mice showing that 95% of cortical astrocytes identified by S100β are recombined, whereas only 1.3% of neurons are recombined. Because individual floxed alleles possess distinct recombination efficiencies, and because we cannot rule out the possibility that a greater proportion of neurons might be recombined in mGFAP Smo CKO mice, we also demonstrated that a broader deletion of Smo using a neuron-specific Cre driver (CamKinaseIIα) does not produce the same phenotypes. We now highlight these data more clearly in Figure 2, and provide further discussion of these points.

2) In light of the Harwell et al., study which showed that pyramidal neurons utilize Shh signaling to regulate dendrite growth and spines in Layer V neurons, and the fact that Smo deletion with mGFAP-Cre is not exclusive to astrocytes, the authors should consider softening their conclusions. Given the limited recombination in other cell types observed with the mGFAP-Cre line, it is likely that the effects observed are related to loss of Smo in astrocytes. However, it is still possible that loss of Smo in other cells contributes to the observed changes. Also, the authors should soften their conclusion that Shh from neurons is mediating the observed changes given that this aspect was not experimentally tested. Regardless of where Shh comes from, it would not alter the conclusion that Shh-dependent signaling in astrocytes alters dendritic spine dynamics and neuronal excitability.

We appreciate the reviewers’ caution. We agree that we cannot rule out Shh from non-neuronal sources, and have now softened this point, as suggested. We have also provided further discussion about the potential contribution of other cell types to our observed phenotypes.

3) The authors should improve illustration of the spine development changes. It is difficult to determine what are spines in Figure 2A and B. This evidence needs to be substantially improved and they should provide representative examples for all of the time points and all of the areas sampled in this figure. The cKO dendrite in Figure 2 seems to be substantially thicker than the WT control. Is this specific to this image, or is this consistent across neurons? If this is a phenotype, please quantify. If not, please choose a different representative dendrite. In Figure 3, it is unclear whether n should refer to spines or to animals. As animals have been subjected to different surgeries, one could argue that n should refer to animals. Please provide further justification for the determination of n in these experiments. The authors should provide additional details about how spines were analyzed and the effectiveness of the custom software in tracking spines should be demonstrated in supplementary material.

We have now provided improved illustration of dendrites and spines, as suggested. Our initial images were derived from brightfield microscopy, using DIC optics. Although these images failed to capture what can reliably be seen on the microscope, we aimed to illustrate our method for performing the spine density analysis in fixed tissues, as described. We now provide maximum projection images from GFP fluorescence captured by confocal microscopy.

We thank the reviewer for pointing out the apparent difference in thickness in the dendrites CKO and wild type. We did not observe any correlation between thickness of the dendrite and genotype in our original analysis, and this was confirmed again in the process of selecting new representative dendrites/images for Figure 2, as described above.

In Figure 3, our n’s were originally reported as cell number. However we reanalyzed the data using animal number, as suggested, and updated Figure 3 accordingly.

We now provide additional details on spine analysis in the Materials and methods, as suggested. Briefly, individual protrusions identified in two z-stacks from the same mouse collected over a defined time interval were analyzed side by side using ImageJ. The presence or absence of individual protrusions was marked using ImageJ’s Cell Counter Plugin, and the turnover ratio was calculated, as described. Analysis was performed by trained observers, blind to genotype. For longitudinal analysis of spine lifetime, the individual features identified in our manual ImageJ analysis were then imported into custom MATLAB-based code, designed to identify corresponding images of previously identified spines, from z-stacks collected at time points beyond the initial 2 day or 7 day period. Corresponding images of an individual spine, across all time points collected are displayed, and the observer scores its presence or absence across all time points. This procedure is now described more fully in Materials and methods, and we also include an additional figure depicting the custom software (Figure 3—figure supplement 2).

4) The results indicate that there are fewer filopodia in juvenile Smo-cKO animals. The authors also cite references supporting a strong correlation between dendritic spine morphology and maturity. To establish whether there are simply fewer filopodial extensions in the Smo-cKO animals, or whether the dendritic spines themselves are more mature, the authors should classify spine morphology as filopodia/intermediate/mushroom-shaped.

We thank the reviewer for this insightful suggestion. We classified the morphology of spines into mature (mushroom-shaped) and intermediate, as suggested. Protrusions exhibiting a bright, bulbous head were scored as mushroom, thin protrusions exhibiting a relatively dim, small head, or no head, were scored as intermediate. Several examples are provided in Figure 3—figure supplement 1. Although WT and CKO mice showed comparable levels of mature spines in juvenile mice, CKO mice showed a relatively smaller fraction of mature spines in adults, compared to their WT controls. A relatively higher proportion of spines were identified as intermediate in juvenile CKO mice compared to controls. While these results were not statistically significant, these data suggest that in CKO mice, a larger fraction of protrusions are mature, or undergoing maturation, compared to WT mice. Because our initial analysis of spine stability did not distinguish between mushroom or intermediate spines, and because our longitudinal analysis of spine stability showed increased survival of spines in CKO mice, this suggests that these intermediate protrusions reflect transitional morphologies as spines undergo maturation and become more stable. These data are shown in Figures 3 and Figure 3—figure supplement 1.

5) The authors provide evidence that Shh signaling in astrocytes regulates neuronal excitability by controlling Kir4.1 expression in astrocytes. The data presented only show a correlation between astrocytes changes (decrease in Kir4.1 expression, increase in GFAP and hypertrophy) and neuronal hyperexcitability. The studies would be more impactful if the authors were to explore whether neuronal excitability was directly related to Kir4.1 downregulation, by assessing whether block of Kir4.1 by Ba^+^ or desipramine in WT neurons elicits a similar increase in excitability.

We agree that specific block of Kir4.1 channels are an important experiment that would enable us to point more directly to Kir4.1 as a molecular mechanism underlying our observed phenotypes. However, neither Ba^+^ nor desiprimine are specific for Kir4.1, and both target other inward rectifying K^+^ channels or adrenergic receptors, respectively, in neurons. We examined the response of cortical neurons in WT slices to desiprimine, and found that at low current injections, the response of neurons is higher in the presence of drug, compared to before drug. However at higher currents, neurons failed to exhibit continued stimulus-dependent responses in response to drug and instead, plateau. While these data partially phenocopies our original result in CKO slices, these data do not necessarily provide more direct evidence for Kir4.1. We have now softened our conclusion that loss of Kir4.1 mediates neuronal hyperexcitability in Smo CKO mice and provided additional discussion of alternative mechanisms underlying this phenotype.